behaviour/computational biology/theoretical biology

mathematical epidemiology, game theory, behaviour, transmission drivers

**Author for correspondence:**
Jaime Cascante-Vega
e-mail: je.cascante10@uniandes.edu.co

[†]These authors contributed equally to this study.

Electronic supplementary material is available online https://doi.org/10.6084/m9.figshare.c.5787146.

# How disease risk awareness modulates transmission: coupling infectious disease models with behavioural dynamics

Jaime Cascante-Vega[1,2,†], Samuel Torres-Florez[1,†], Juan Cordovez[1] and Mauricio Santos-Vega[1]

[1]Departamento de Ingeniería Biomédica, Grupo de Investigación en Biología Matemática y Computacional, Universidad de los Andes, Bogotá D.C., Colombia
[2]Department of Environmental Health Sciences, Mailman School of Public Health, Columbia University, New York, NY 10032, USA

 JC-V, 0000-0002-3958-6665; ST-F, 0000-0002-0590-4282;
MS-V, 0000-0002-8130-6014

Epidemiological models often assume that individuals do not change their behaviour or that those aspects are implicitly incorporated in parameters in the models. Typically, these assumptions are included in the contact rate between infectious and susceptible individuals. However, adaptive behaviours are expected to emerge and play an important role in the transmission dynamics across populations. Here, we propose a theoretical framework to couple transmission dynamics with behavioural dynamics due to infection awareness. We modelled the dynamics of social behaviour using a game theory framework, which is then coupled with an epidemiological model that captures the disease dynamics by assuming that individuals are aware of the actual epidemiological state to reduce their contacts. Results from the mechanistic model show that as individuals increase their awareness, the steady-state value of the final fraction of infected individuals in a susceptible-infected-susceptible (SIS) model decreases. We also incorporate theoretical contact networks, having the awareness parameter dependent on global or local contacts. Results show that even when individuals increase their awareness of the disease, the spatial structure itself defines the steady state.

## 1. Introduction

Traditional infectious disease transmission models allocate the population into compartments that capture different disease

states and aim to parametrize the rates of transition between those states in a manner that reflects the underlying biology of the disease [1,2]. Numerous factors influence transmission and are important to consider or model directly in epidemiological models. One of those is the understanding of the effect of an individual's behaviour on the disease population dynamics, which has recently been highlighted as a response to reduce contact rates and therefore pathogen transmission across populations [3]. Typically, as the disease spreads in a population, individual behaviour can change and therefore infection probability can be reduced or amplified [4–7]. Recently, changes in individual contact rates driven by changes in the awareness on the epidemic or the state of the infectious disease have been discussed and explicitly modelled [8,9]. Although these models do not incorporate an explicit mechanism by which individuals could modify their behaviour, they have demonstrated how behavioural aspects play an essential role in disease dynamics. Over time, behaviour is expected to vary as population-level disease awareness is modulated by increases in risk and the proportion of the population that has been infected (e.g. risk awareness might be highest near the peak of the epidemic) [9]. However, the impact of behaviour–time dynamics in controlling transmission is probably intertwined with other variables that directly impact transmission, and can be estimated in a time-variable contact rate. Recently, COVID-19 has highlighted the importance of sustained social distancing to reduce infection risk within the population so health systems capacity is not saturated [3–5]. As epidemics unfold, individuals amass information provided by public health institutions concerning the status of an epidemic or are strongly influenced by beliefs of the disease in their population. This heterogeneity in the levels of information gathered in certain communities could modulate the level of adherence to certain interventions. A classic example is the Ebola outbreak in Sierra Leone, where risk communication played an essential role in controlling transmission due to high infection probability given contact with an infectious individual of this disease [10].

Mechanistic models that aim to explain the dynamics of human behaviour often rely on a game theory framework, where behaviour is represented by the strategy that an individual adopts in a decision-making process [11–14]. However, most coupled epidemiological-game theory models assume static games and therefore do not capture the dynamics of how individuals change their behaviour over time, as an epidemic unfolds [15,16]. By contrast, evolutionary game theory has been developed precisely to model decision-making as a time-dependent process. By using simple assumptions about how different behavioural strategies generate a payoff (i.e. an advantage in evolutionary fitness), they model decisions across individuals and therefore how the adoption of different behaviours might affect the disease dynamics. Epidemiological models have also been proposed for studying information diffusion on adaptive social networks [17,18] and have also been coupled with information dissemination dynamics to link disease transmission [19,20].

Both epidemiological and behavioural dynamics have been studied under space–time couplings resulting in complex dynamics depending on the contact or social network where interactions between individuals take place [7,21]. This highlights the importance of understanding both coupled and dynamical models for defined social networks where contacts represent interactions between individuals (or nodes). This would provide greater insight into how network topology may modulate disease awareness and ultimately how an epidemic may unfold [22,23].

In this work, we introduce a novel framework for coupling both epidemiological and behavioural models. Our framework consists of a traditional epidemiological model [2] and a replicator dynamics model [11,12]. By combining these two approaches we were able to dynamically couple individuals' behaviours with transmission intensity. Our work sheds light on the interaction between social and behavioural dynamics affecting the epidemiology of the disease. Specifically, our results show that even when individuals increase the disease awareness, the contact network itself defines the steady-state solution of the system. In addition, more connected networks (networks with random or constant degree distributions) result in a population with no change in their behaviour. Our work shows that explicitly incorporating responsive behavioural dynamics can significantly change the predicted course of an epidemic and highlights the importance of accounting for this source of variation. We also studied the case where an individual's behaviour is non-elastic; then there is no change in their behaviours during the epidemic.

## 2. Material and methods

We study these coupled epidemiological and social dynamics using two different models: (i) We couple two ordinary differential equations systems (ODEs), one to describe the state transitions in

the epidemiological model and the other to describe the replicator dynamics. The epidemiological model is a susceptible–infected–susceptible (SIS) compartmental model, where susceptible individuals in the population are those who are immunologically able to acquire a novel pathogen; infected are those who have acquired the pathogen and are infectious. We assume that the force of infection (FOI) is inversely proportional to the fraction of individuals in the population who cooperate $c$ (here cooperation is understood as the strategy that reduces epidemiological risk but does not necessarily provide the highest payoff). Then, following the rationale that individuals who do not cooperate might have a higher FOI, we discount the defector (non-cooperator) payoff by the fraction of infected individuals $I/N$ times a parameter that we call the awareness $\sigma$. (ii) Our other approach was to couple the epidemiological dynamics in an explicit contact network $\mathcal{G}$, where nodes represent the individuals and the edges represent the contacts [6]. We then sequentially update the replicator equation within the network. Similar to the coupled ODE models, we assume the FOI is inversely proportional to the number of cooperating individuals; however, as the network model considers contacts of an individual we attempt to model individual access to the state of infectious individuals in two settings that we call global and local information. This follows the rationale that individuals can be informed of the disease at multiple spatial scales; they may be aware of the proportion of infected individuals within the entire population (global information), or they may know the state of infection for their neighbours defined as the individuals that the individual have contact with (local information). We study the dynamics in three types of theoretical constructed networks: scale-free, random and grid networks. We study the change in contact rates as a function of the fraction of individuals in the population who cooperate $c$ using five different functional forms as is shown in electronic supplementary material, figure S4. We also study the effect of having a fraction of individuals in the population whose behaviour is not elastic, and therefore never change their behaviour.

## 2.1. Mechanistic ODEs model

We consider an SIS (susceptible–infected–susceptible) model (equation (2.4)) which is a variation of the Kermack & McKendrick model for diseases without or with short-term immunity, which forms the basis of almost all the communicable disease models studied. In our SIS model, the population is divided into two classes, where susceptible ($S$) can be infected by those already infected ($I$) and subsequently become susceptible again due to lack of immunity [2]. We then model strategic interactions between individuals using the replicator dynamics (RD), a concept from evolutionary game theory where population's behavioural traits are described using biologically inspired operations such as natural selection [11,12,24]. Under these dynamics, the percentage growth rate of cooperators $\dot{c}/c$ is equal to the excess of payoff with respect to the populations payoff. We set $f_c$ and $f_n$ as the fitness of cooperators and non-cooperators, respectively. Therefore, the average fitness can be obtained as $\bar{f} = c \cdot f_c + n \cdot f_n$.

and
$$\left.\begin{array}{l} \dfrac{\mathrm{d}c}{\mathrm{d}t} = c(f_c - \bar{f}) \\[2mm] \dfrac{\mathrm{d}n}{\mathrm{d}t} = n(f_n - \bar{f}). \end{array}\right\}$$
(2.1)

We then model the payoff using the prisoners dilemma (PD), an archetypical model displaying the conflict between selfishness and public good [25]. It is a $2 \times 2$ game with two strategies, cooperate or non-cooperate. The payoff modelling pair meetings is defined in equation (2.2). Here we set $S = 0.5$ and $T = -0.5$ as in the PD [25]. Note that the row referring to the payoff that receives non-cooperator individuals is discounted by the awareness of the disease $g(\sigma, I)$.

$$A = \begin{array}{c|cc} & C & N \\ \hline C & 1 & S \\ N & T - g(\sigma, I) & -g(\sigma, I) \end{array}$$
(2.2)

Then the fitness of each strategy $f_c$ and $f_n$ can be written as shown in equation (2.3). Note that the fitness is modelled as the expected payoff given current frequency of cooperators $c$ and non-cooperators $n$. Similarly, the average fitness $\bar{f}$ is then the expected given the fitness of each one of

the populations.

$$\left.\begin{aligned} f_c &= c \cdot 1 + n \cdot S \\ f_n &= c \cdot (T - g(\sigma, I)) + n \cdot -g(\sigma, I) \\ \bar{f} &= c \cdot f_c + n \cdot f_n \end{aligned}\right\} \tag{2.3}$$

and

In the typical PD, the non-cooperator strategy is the dominant strategy. However, the payoff, if both players defect, is less than the payoff if both cooperate, resulting in a social dilemma. Here we assume cooperators' payoff remains the same no matter the FOI in the population, as their behaviour might not be perceived less if they follow rules to reduce disease spread. By contrast, non-cooperator payoffs are reduced in a manner proportional to the population awareness $\sigma$ and the fraction of infected individuals in the population $I/N$. This accounts for the assumption that payoff is reduced proportional to the FOI as it is coupled with the number of infected individuals in the population. We then assume individuals' strategy affects transmission rate $\beta$ as a function of the fraction of cooperators in the population $c$, then $\beta(c) = f(c)$. As our model attempts to study dynamics in a short time period (scales at which individuals' behaviour might change), we do not attempt to include birth or death process in the model. Equation (2.4) shows the coupled dynamics resulting in the epi-social model. In the electronic supplementary material is included the algebraic reduction of the two-dimensional replicator dynamics describing the change of cooperators and non-cooperators to the equation describing the change of cooperators in line $dc/dt$.

$$\left.\begin{aligned} \frac{dS}{dt} &= -\beta(c)\frac{SI}{N} + \gamma I \\ \frac{dI}{dt} &= \beta(c)\frac{SI}{N} - \gamma I \\ \frac{dc}{dt} &= c(1-c)(f_c - f_n). \end{aligned}\right\} \tag{2.4}$$

and

The matrix payoff governing the dynamics of the game is then given by equation (2.2). We assume non-cooperators' payoff is discounted by the awareness of the disease given the current epidemic state $g(\sigma, I)$. The level of awareness that a population exhibits is named $\sigma$, and the awareness population given the current epidemic state is defined by equation (2.5). Finally, the transmission probability is assumed to be inversely proportional to the fraction of cooperators in the population. We explored different functional forms to describe the relationship between $\beta(c)$ and $c$. First, for simplicity we only assume contact rate decays exponentially with the cooperator fraction $c$ as is shown in equation (2.6). This functional response has the advantage that it only needs to be parametrized with the nominal contact rate named $\beta_{max}$ in a population where behaviour is assumed to not impact transmission at all (i.e. $\sigma = 0$, an unconscious population). We then explore four more functional forms depicted in electronic supplementary material, figure S4, where the rate of change in the contact rate $\beta(c)$ varies given a change in the fraction of cooperators $c$. Here defined the minimum value for all the functional responses as $\beta_{min} = \beta_{max}\exp(-1)$. Choosing a minimum contact rate follows the rationale that assuming all individuals cooperate, $c = 1$, the exponential functional response equation (2.6) results in the minimum value of the transmission probability we consider

$$g(\sigma, I) = \frac{\sigma \cdot I}{N}, \quad \sigma \in [0, 1] \tag{2.5}$$

and

$$\beta(c) = \beta_{max} \exp(-c). \tag{2.6}$$

## 2.2. Network model

Interactions between individuals (i.e. explicit contacts) are modelled assuming mass-action law. We therefore assume homogeneous contacts between individuals regardless of their state of the disease, susceptible or infected. However, considering explicit social networks give additional insight about the role of contacts, usually studied within theoretical constructed networks [8,26]. We implemented an SIS model on a network assuming now susceptible individuals get infected by their infected neighbours (individuals with common edges) by an infection probability $\beta_i$ and recover from the disease with the rate $\gamma$, same as used in the ODEs model equation (2.4). We integrate the models

using a daily time-step of $\Delta_t = 1$. Then each individual $i$ has a state in the disease named susceptible $S$ or infected $I$ and the transition between those states is as described by equation (2.7)

$$\left.\begin{array}{l} P(S \to I) = \beta_i \\ P(I \to S) = \Delta_t \cdot \gamma. \end{array}\right\} \tag{2.7}$$

and

To extend the behavioural dynamics on a network, we again use the replicator dynamics at individual level (or imitation dynamics) update rule given a payoff matrix as the one described by equation (2.2). We assume that individuals tend to adopt the strategy of more successful neighbours, where success is measured with the payoff as described before, more successful individuals correspond to the ones with greatest payoff in a time step of the simulation. Then successful individuals are pushed to both balance the social dilemma between cooperate or non-cooperate given the current state of the epidemic. Similar to the replicator dynamics, mean field approximation assumes a two-strategy game whose update rule is described by equation (2.8), where the probability of changing from the strategy used by the player $j$ to the strategy used by the player $i$ is given by the differences of payoffs of each player $u_j$ and $u_i$, respectively, and player irrationality $K$, which we set to 0.5.

$$P_{i \leftarrow j} = \frac{1}{1 + \exp\left(-(u_j - u_i)/K\right)}. \tag{2.8}$$

Therefore, the payoff of player $i$ comes from playing the same symmetric two-player game defined by the used $2 \times 2$ payoff matrix. In this case, they total income can be expressed as equation (2.9)

$$u_i = \sum_{j \in \mathcal{N}_i} s_i A s_j, \tag{2.9}$$

where $s_i$ is the unit vector describing the pure strategy played by the individual $i$ and the summation runs on the individual neighbours $j \in \mathcal{N}_i$ defined by the network structure [21,27]. The infection rule is given by the infection probability of each node $\beta_i$, which similarly as the ODEs model this node infection probability will be inversely proportional to the number of cooperating individuals in her neighbourhood $\beta_i = \beta(c_{\mathcal{N}_i})$. Again for simplicity we start assuming that infection probability decays exponentially with the fraction of cooperators in the neighbours as is shown in the equation (2.10). This functional response again can only be parametrized using the nominal infection probability named $\beta_{\max}$. We then explore four more shapes shown in electronic supplementary material, figure S4, that vary the rate of change of the contact rate $\beta(c)$ given a change in the fraction of cooperators $c$. We fixed the minimum value for all the functional responses as $\beta_{\min} = \beta_{\max}\exp(-1)$ as explained in previous section.

$$\beta_i = \beta_{\max} \cdot \exp(-c). \tag{2.10}$$

The social dynamics will consequently depend on the current state of the disease. We discount the payoff of non-cooperator individuals by the awareness parameter $\sigma$ and the current state of the disease $g(\sigma, I)$. However for modelling the effect on local versus global spread of the information we assume this discount factor will depend on the state of the disease in neighbouring nodes (local information) or in the whole population (global information) as described by equation (2.11). This approximation follows the rationality that disease dynamics are governed by local outbreaks (infection among nodes and their neighbours) as is shown in figure 1b,c.

$$\left.\begin{array}{l} g(\sigma, I) = \sigma \dfrac{I}{N} \quad \text{global information} \\ g(\sigma, I) = \sigma \dfrac{1}{|\mathcal{N}_i|} \displaystyle\sum_{i \in \mathcal{N}_i} I_i \quad \text{local information.} \end{array}\right\} \tag{2.11}$$

and

## 2.3. Simulation and steady-state analysis

For both models, we assume individuals on average are infectious for 7 days and therefore the recovery rate is $\gamma = 1/7$. We first study the steady state of both ODEs and network models and characterize the dynamics of the system in the steady state. We calculated the fraction of infected individuals $\bar{I} = I/N$ and the fraction of cooperators $c$. Given that the network model is stochastic and sensitive to the number of infected nodes in the network at the beginning, we ran 100 simulations in a 2000-node network for $T = 150$ days after the first infection was seeded. Additionally, to compute the steady state

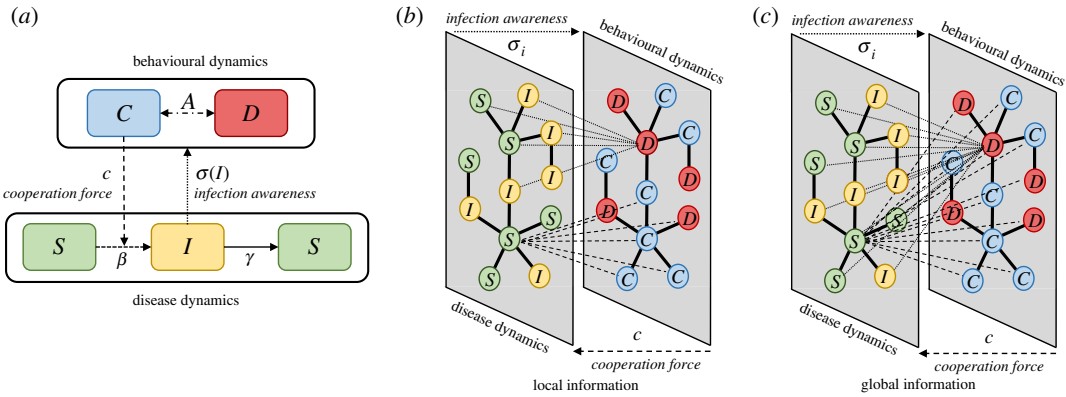

**Figure 1.** Coupling disease and behavioural dynamics. (*a*) Schematics for a deterministic framework which is replicated in (*b,c*) aiming for an individual-based approach. (*b,c*) Schematics for the probabilistic implementation on a multi-layer network. (*b*) Local information is only transmitted through adjacent nodes $\omega_i$. For the behavioural dynamics, an individual's behavioural strategies are only affected by their awareness $\sigma$ of the disease state of the neighbouring nodes ($\omega$) at time *t*. Similarly, the infection probability $\beta_i$ of any node, depends only on the behaviour of their neighbours, which is related to the number of cooperating individuals *c*. By contrast, (*c*) describes global information transmission in the population. Here, individuals base their strategies on the awareness $\sigma_i$ of the infection state for the entire network. Similarly, the infection probability $\beta_i$ of any node will depend on the cooperation *c* of all the population

$\bar{x}$ of variable $x(t)$ we averaged in the last week of the simulation as shown in equation (2.12) to reduce the effect of the stochastic behaviour in the network simulation.

$$\bar{x} = \frac{1}{7}\sum_{i=0}^{6} x(T-i). \tag{2.12}$$

To seed the initial conditions, we set the initial condition as 50% of the population as cooperators. Note that, as the non-cooperator strategy is the dominant strategy in the payoff matrix, this model is not sensible to the initial conditions. This seeding follows the rationale that a randomly determined fraction of the population cooperates voluntarily toward reducing their risk of getting infected with the disease. We then vary the contact rate or probability of infection $\beta$ and $\beta_i$, respectively, for ODEs and network model between 0.3 and 1. In the results, rather than presenting the contact rate $\beta$, we use the basic reproductive number $R_0 = \beta/\gamma$ of the disease, as it is a quantity one can relate easily with an specific disease and itself has a meaning in the epidemiological context. This basic reproductive number $R_0$ corresponds to the expected number of infected individuals given an initial one in a completely susceptible or free of the disease population. This results in varying the $R_0$ between 0.7 and 7 with the specified recovery rate $\gamma = 1/7$.

We then choose the nominal transmission probability values max to result in a basic reproductive number $R_0$ in three transmission regimes $R_0 \in 2.2, 4.2, 6.3$ that represent normal, high and super high transmission settings. Different pathogens therefore are mapped in these levels of transmission. For example, SARS $R_0$ was estimated to be in the range 2.2–3.6 (normal transmission) [28], SARS-CoV-2 $R_0$ estimates presented early in the COVID-19 pandemic present mean values from 2.5 to 6.1, but reconciled estimates accounting for differences in growth rates and generation intervals present values from 2.3 to 4.8 (high transmission) [29]. And finally for measles $R_0$ is generally cited to be in the range 12–18; however, different estimates considering prior population immunity by vaccination estimate it in the range 6–9.5 (super high transmission) [30–32].

Finally, to analyse the role of the heterogeneity in contact network favouring cooperation, we study the temporal dynamics of the model at two scales using three levels of awareness that we call full, partial and half awareness for $\sigma \in \{1, 0.7, 0.5\}$, respectively. These scales correspond to a population scale and a clustered-neighbours scale. This idea follows the rationale that networks exhibit a high clustering coefficient [21] suggesting that in steady-state hubs of nodes might favour cooperation. We therefore analyse the disease dynamics in those clusters. To determine the effect of transmissibility of the disease we consider five values of $R_0 \in \{6.3, 4.2, 3.1, 2.1, 1.5\}$. We do not consider $R_0$ values less than 1. In electronic supplementary material, algorithm 4, we include further information of the clustering algorithm used.

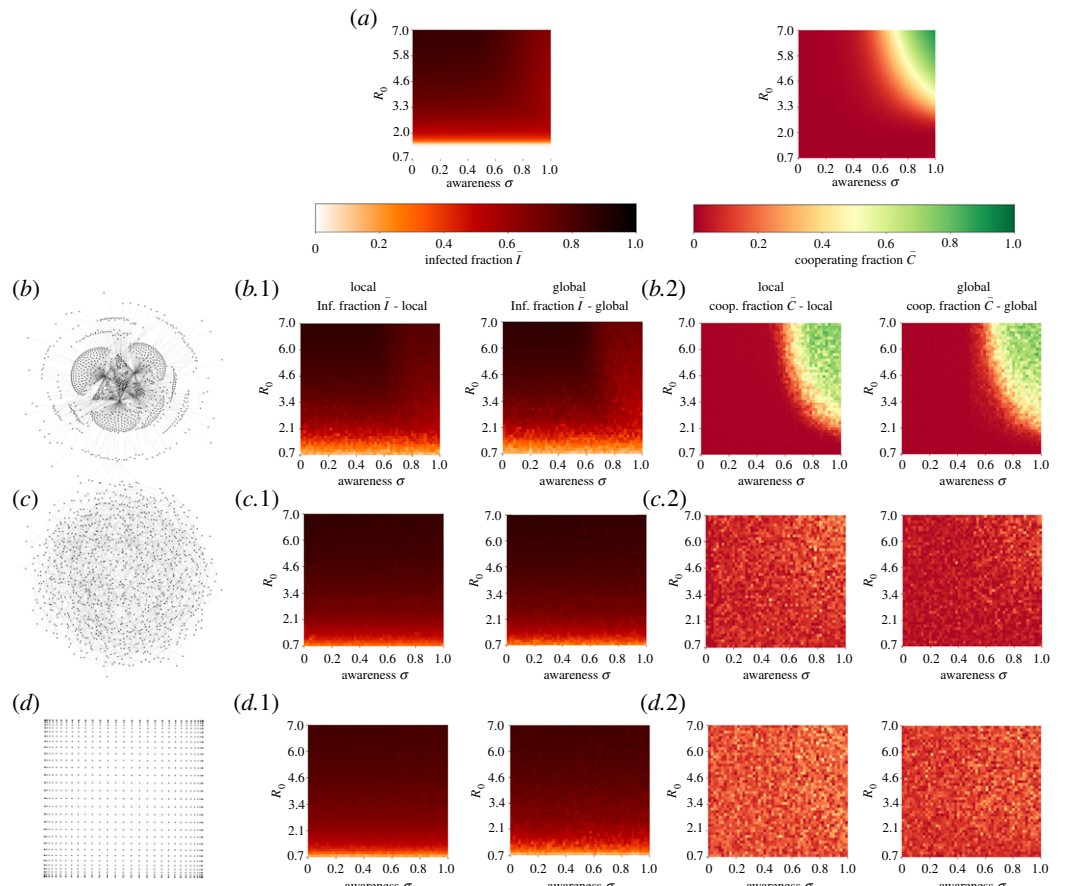

**Figure 2.** Steady-state solution of population infected fraction, and cooperators fraction. The $y$-axis of all heat-maps represents basic reproductive number $R_0$ and the $x$-axis represents awareness level $\sigma$. Local and global simulations are only presented for network models. ($a$) ODEs model results for infected fraction (left) and cooperation fraction (right). ($b$) Network model on scale-free graph for ($b$.1) infected fraction and ($b$.2) cooperation fraction. ($c$) Network model on Watts–Strogatz/small-world graph for ($c$.1) infected fraction and ($c$.2) cooperation fraction. ($d$) Network model on grid graph for ($d$.1) infected fraction and ($d$.2) cooperation fraction.

# 3. Results

Our results focus on understanding the role of behaviour modelled as the replicator dynamics present in the ODEs model and the imitation update rule in the network model. Figure 2 presents the steady-state solution for all the models used with the exponential shape shown in electronic supplementary material, figure S4. Figure 2$a$ shows steady states for the ODEs system using the SIS and replicator dynamics-coupled model, figures on the left correspond to the final fraction of infected individuals running the model with different values of $R_0$ and awareness $\sigma$. Darker red corresponds to a higher fraction of infected individuals in the steady state. Similarly, figures on the right show the final fraction of cooperators in the model where green corresponds to a higher final fraction of cooperators in the steady state. Electronic supplementary material, figure S5 presents the steady-state solution for all the different shapes between the transmission rate $\beta(c)$ and the cooperators fraction $c$ for the ODEs model. Similarly, electronic supplementary material, figures S6 and S7 shows the steady-state solutions for the scale-free network and all the shapes between $\beta(c)$ and $c$ considered using the global and local information scheme, respectively.

Real contact networks are usually highly heterogeneous, and often scale-free as well as highly clustered. Figure 2$b$–$d$ shows the median steady-state dynamics after running the model 100 iterations for scale-free, Watts–Strogatz or small-world and grid graphs, respectively. Figure 2$b$.1–$d$.1 shows steady-state fraction of infected individuals $\bar{I} = I/N$. The heat-maps on the left correspond to model runs in the global scenario, where awareness discount factor considers all the infected individuals in the population, while heat-maps on the right correspond to runs of the model in a local

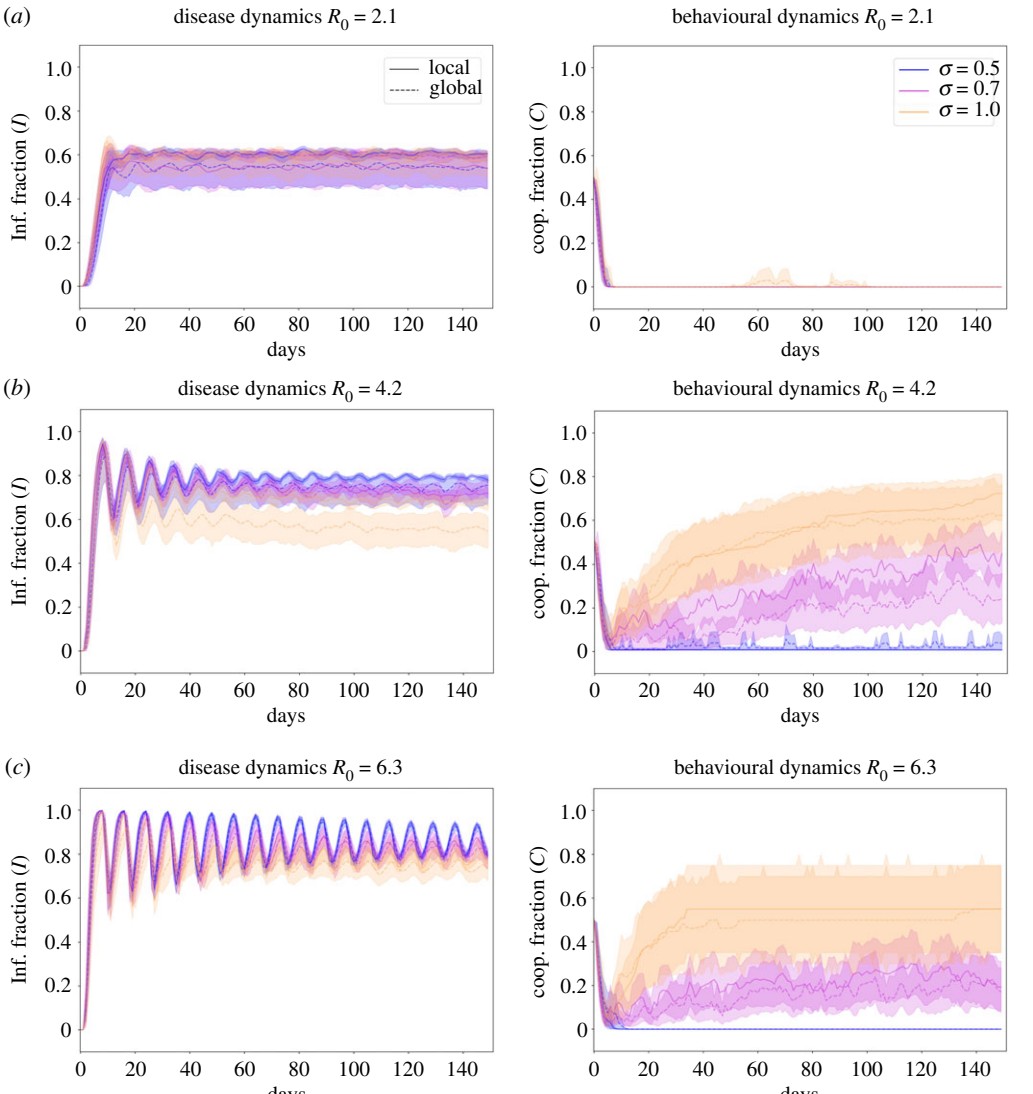

**Figure 3.** Temporal evolution for different levels of awareness $\sigma \in \{0.5, 0.7, 1.0\}$ and three levels of transmission (a) $R_0 = 6.3$, (b) $R_0 = 4.2$, (c) $R_0 = 2.1$. All lines represent the median of the 100 simulations and ribbons represent the 90% quantiles. Left column shows the infected fraction in time $I(t)/N$ and the right column shows the cooperators fraction $c$. The solid line corresponds to running the network model in the global setting while dashed line corresponds to the local setting. We simulate the model for $T = 150$ days after the first infection was seeded.

scenario, where awareness discount factor depends on each individual proportional to the infected fraction in their neighbourhood. Similarly, figure 2b.2–d.2 shows the steady-state fraction of cooperating individuals $\bar{c}$. Left and right columns correspond to running the model in the global and local setting; again these settings affect the awareness discount factor given the current state of the disease.

While steady states characterize the overall effect of transmission and social parameters $R_0$ and $\sigma$, respectively, temporal dynamics of both infected and cooperators exhibit interesting changing behaviour as transmissibility increased, leading the dynamical system to their steady state faster [2,33]. Then suggesting behaviour dynamics, i.e. cooperation will also be faster as a response of reducing the transmission of the disease. We then study time dynamics under three levels of awareness named full, partial and medium awareness as we indicated in the Material and methods section and three levels of transmission that relates pathogens from normal transmission approximately 2 or highly transmitted approximately 6. Specifically, we use $R_0 \in \{2.1, 4.2, 6.3\}$ as discussed in Material and methods section; this is shown in figure 3. Each row represents a different transmission $R_0$ value and colours represent a given level of awareness. We extended the results for the different functional forms between $\beta(c)$ and the cooperators fraction $c$. These results are depicted in electronic supplementary material, figures S8 and S9 for the infected individuals fraction and cooperator

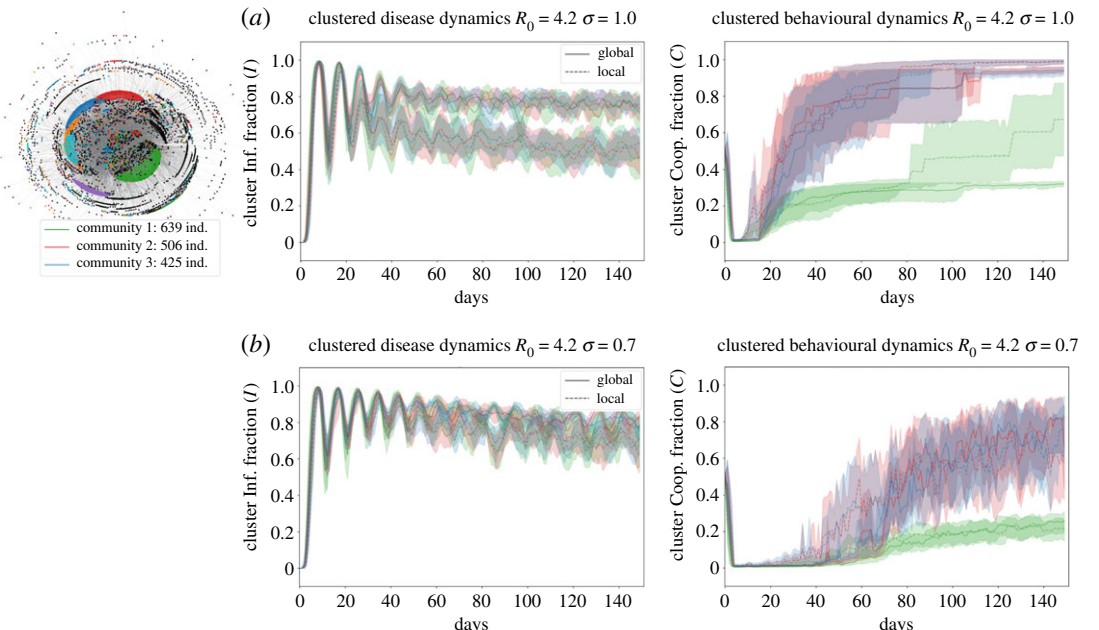

**Figure 4.** Temporal evolution for different levels of awareness. (a) High awareness $\sigma = 1.0$ and (b) medium awareness $\sigma = 0.7$ as described in Material and methods section. Each colour represents a hub (set of nodes) identified with the community clustering algorithm as described in the electronic supplementary material. We only studied dynamics in the three hubs with more nodes. All lines represent the median of the 100 simulations and ribbons represent the 90% quantiles. Left column shows the infected fraction in time $I(t)/N$, and the right column shows the cooperators fraction $c$. The solid line corresponds to running the network model in the global setting while dashed line corresponds to the local setting. We simulate the model for $T = 150$ days after the first infection was seeded.

individuals fraction, respectively. We include the same results for the temporal dynamics on scale-free networks shown in electronic supplementary material, figures S10 and S11 for the infected individuals fraction and cooperator individuals fraction, respectively.

Then we interrogate about the impact of having a fixed fraction of individuals in time have a non-elastic behaviour and therefore never or always cooperate. We simulated the system assuming that a random 10%, 25% and 50% of individuals do not cooperate (or defect) and cooperate for all the simulation time. Electronic supplementary material, figures S12 and S13 show the mean time dynamics for the infected individuals fractions and cooperator individuals fraction, respectively, in the global information scheme and electronic supplementary material, figures S14 and S15 for the local information scheme.

It has been shown that evolutionary dynamics on scale-free networks might favour cooperation (sustain it in time) in highly connected nodes [21]. Therefore, we hypothesize that both infected fraction and cooperators fraction might change across scale-free network clusters. In consequence, we search for clusters using a node clustering algorithm on the network and plot the fraction of infected individuals; $(\bar{I}_i = (1/\mathcal{G}_i) \sum_{j \in \mathcal{G}_i} \mathbb{I}(j = I)$ where $\mathcal{G}_i$ correspond to the set of nodes identified in the $i$th cluster and $\mathbb{I}(j = I) = 1$ if the $j$th node is infected). Similarly, we computed the cooperator fractions in these clusters $\bar{c}_i = (1/\mathcal{G}_i) \sum_{j \in \mathcal{G}_i} \mathbb{I}(j = c)$, where $\mathcal{G}_i$ corresponds to the set of nodes identified in the $i$th cluster and $\mathbb{I}(j = c) = 1$ if the $j$th node is cooperating. Figure 4 shows the temporal dynamics for $T = 150$ days after the seeded index case. The clusters computed are also shown at top left side of figure 4; note that we only consider the three biggest hubs (top three clusters detected with more nodes). We show the temporal dynamics in figure 4a,b, here the row (a) correspond to a maximum level of awareness and the row (b) to a medium level of awareness, we do not show results for low level of awareness as is shown in figure 2b.2 cooperation does not occur at that level. The middle and right columns show final infected and cooperators fraction in each cluster; the lines indicate the setting in which we run the simulation, with the solid line corresponding to the global setting and the dashed line corresponding to the local setting.

Finally, we study the temporal dynamics on a small-world and grid network shown in electronic supplementary material, SI figure 3. The temporal dynamics for the ODE model are shown in electronic supplementary material, figure S2. We do not attempt to study clustering structure in the

small-world or grid networks as their degree distribution is uniform and therefore the hub might correspond to the whole network. We restrict the analysis of hub dynamics to the scale-free graph.

## 4. Discussion

Social dynamics play an essential role in the evolution of epidemics and the unfolding of a disease across a population [6,8]. Specifically, awareness of the state of the epidemic or disease can result in different shapes or patterns in the epidemic curve [9,34]. The latter implies that the structure of population also plays an important role in both the evolution of disease spread and social dynamics [21]. Real heterogeneous networks such as the scale-free networks are used to resemble real-world ones and have been shown to favour cooperation in a social dilemma [21]. We developed a theoretical framework that couples two models typically used for modelling the spread of a pathogen in a population, the SIS model, and for modelling the evolution of cooperation in social dilemmas [2,25]. To our knowledge, it is the first infectious disease model to couple social behaviour with disease dynamics, using replicator dynamics to examine how awareness of the disease state within a social network may regulate the adoption of protective behaviours. Particularly, we assume social dynamics follow a probability distribution and link the cooperator philosophy to actions toward reducing transmission in the population by assuming contact rate or infection probabilities $\beta$ is a decreasing function of the cooperators in the population.

We studied the steady-state dynamics of the system as shown in figure 2 and found that ODEs and network models in a scale-free have similar if no identical steady-state dynamics as shown in figure 2a,b. As awareness levels $\sigma$ in the population increase, the final fraction of cooperators increase to nearly 100%. Interestingly, these dynamics also exhibit a strong relation with disease transmissibility measured here with $R_0$. We found that for low levels of transmission $R_0 < 2.1$ no matter what level of awareness is in the population, no one cooperates (figure 2a right and figure 2b.2), and therefore the final fraction of infected individuals is what the epidemiological system steady state dictates (figure 2a left and figure 2b.1). These results also confirm our hypothesis that by imposing a discount factor on the human behaviour dynamics (here fraction of infected individuals) on the payoff of non-cooperators, the steady state of this dynamics results in a high fraction of the population cooperating towards reducing the disease transmission. This in turn results in a lower fraction of infected individuals when the system reached a steady state.

We extended our results considering multiple shapes in the relation between $\beta(c)$ and $c$ (electronic supplementary material, figures S5–S7). Our results consistently found that the steady states of infected and cooperators fraction in the ODEs model and both local and global information scheme network model is determined by this shape. We intentionally present our results from left to right sorting the shape so that a change in the cooperator fraction $\Delta c$ results in a smaller change in the contact rate $\beta(c)$. This, therefore, impacts the steady-state dynamics similarly where the concave 2 shape results in a higher fraction of cooperators in the steady state for certain values of awareness $\sigma$. However, the steady state of the epidemiological models converge asymptotically to the same infected fraction. This confirms that the equilibrium of the infected individuals is not affected by the shape between the contact rate and the cooperator's fraction. These results can also be seen in the time dynamics shown in electronic supplementary material, figures S8–S11.

Comparing the final fraction of cooperators in the small-world versus the grid graph (figure 2c.2 and figure 2d.2, respectively), we observed that despite the transmission strength $R_0$ or the awareness $\sigma$ nearly 25% of the population cooperates (yellow colour in the heat-map). This is confirmed visually by looking at the steady state in figure 3 and electronic supplementary material, figure S3. We hypothesize that these dynamics emerge by the natural structure of the contact network, which naturally pushes the system towards cooperation and sustains it in time, as has been shown for other real-world networks [21].

We also highlighted the importance of considering individuals who follow a fixed strategy. This effect is illustrated in electronic supplementary material, figures S12 and S13 for the global information scheme, and in electronic supplementary material, figures S14 and S15 for the local information transmission. A higher percentage of fixed cooperators will guarantee that the system remains in a highly cooperative environment despite lower levels of disease awareness and will reduce the epidemic size. However, this effect loses importance when considering high infectious scenarios.

Although, our framework provides a flexible way to dynamically incorporate behaviour in epidemiological models; there are important limitations to highlight. First, the ODEs model, as is usual, uses compartmental models to understand the transmission of pathogens across populations,

assuming homogeneous mixing of the populations leading to a biased FOI when the contacts in the populations are highly heterogeneous. The heterogeneous features in contact patterns are principally caused by both mobility of individuals across communities, and spatial segregation of communities principally caused by socio-economic factors [35]. To account for this characteristic, we implemented the network models. However, these models, as does an agent-based model (ABM) with a specific contact network, need to update each agent state in each time step. In consequence, its running time performance is poor for large populations compared to the ODEs models. Moreover, as epidemics unfold across populations individuals' behaviour changes as addressed in our work by voluntarily reducing their FOI or by imposing directly public health non-pharmaceutical interventions (NPIs). This leads to a huge heterogeneity around interactions of people inside and outside their households, in the community, workplace, etc., therefore varying the contact structure (social network) in time. In consequence, assuming a static contact network might reflect poorly in real-world settings. This, however, can be further improved as an understanding of local social networks and/or data collection about individuals' movement is improved. In addition, our model considers a game-theory framework based on where individuals change their behaviour in pair-wise encounters. Parametrizing these models might be a challenge in the ongoing epidemic due to the huge uncertainty in modelling real-time people's behaviour. Other methods should be explored to account for these dynamic networks.

Our framework accounts for dynamic changes in the behaviour of an individual, diminishing their own risk of infection as well as the risk of those with whom it interacts [36]. To incorporate this in the models, we considered individual-level contact patterns to capture behaviour towards a disease outbreak. This allows an analysis of the disease and behavioural dynamics in complex population structures, and permits explicit demographic predictions that can be studied for public health interventions [37]. We explored the impact of population structure in epidemiological outcomes with different disease and behavioural assumptions set for the $R_0$ and risk awareness.

In general, we have evaluated the effect that communities have on the spread of cooperative strategies towards the mitigation of a disease. We highlight the importance of dense population structures in maintaining cooperative regime, which in turn decreases the epidemic size within the whole population. This crucial characteristic of heterogeneous scale-free graphs is absent in homogeneous scatter and small-world graphs, where this population architecture fails to achieve cooperation in order to counter the spread of an infectious disease.

Data accessibility. Data and relevant code for this research work are stored in GitHub: https://github.com/ChaosDonkey06/DiseaseRiskAwareness_ModulatesTransmission and have been archived within the Zenodo repository: doi:10.5281/zenodo.5525229.

Competing interests. We declare we have no competing interests.

Funding. J.C. and M.S.V. were supported by IDRC-Uniandes grant no. 109582-001.

Acknowledgements. We would like to acknowledge Pallavi Kache, Alejandro Feged-Rivadeneria, Tomas Rodríguez Barraquer and Pablo Cárdenas for their thoughtful comments on the manuscript.

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
