## [Peer Review File · Royal Society Open Science]

Review History

RSOS-210803.R0 (Original submission)

Review form: Reviewer 1

Is the manuscript scientifically sound in its present form?

Yes

Are the interpretations and conclusions justified by the results?

Yes

Is the language acceptable?

Yes

Do you have any ethical concerns with this paper?

No

Have you any concerns about statistical analyses in this paper?

No

Recommendation?

Major revision is needed (please make suggestions in comments)

Comments to the Author(s)

In this paper, a coupling transmission dynamics with behavioral dynamics due to infection awareness is constructed and the dynamics of social behavior is modeled by game theory. The results show that as individuals increase their awareness the steady-state value of the final fraction of infected individuals in SIS model decreases. They extend the results to a spatial framework, incorporating a spatially-defined social network and show that even when individuals increase their awareness of the disease, the spatial structure itself defines the steady-state. The idea of this article is novel. The problems of this paper are as follows:

1. In the introduction of P 2, the ninth line of the first paragraph and the eighth line of the second paragraph, there are some writing mistakes, please check it.
2. In the Introduction part, the authors mentioned some work on mathematical modelling on both epidemiological and behavior change. However, some related work needs to be included: Linking the disease transmission to information dissemination dynamics: An insight from a multi-scale model study, *Journal of Theoretical Biology*, 2021; Markov-based solution for information diffusion on adaptive social networks, *Applied Mathematics and Computation*, 2020; Coupling dynamics of epidemic spreading and information diffusion on complex networks, *Applied Mathematics and Computation*, 2018; Evaluating risks using simulated annealing and Building Information Modeling, *Applied Mathematical Modelling*, 2015.
3. Can the derivation of the equation describing the behavior change be more detailed in the Eq.2.4 of page 4?
4. In the Network Model of P5, the line 7 of the first paragraph, $Pr = 1/\gamma$, should it be $Pr = \gamma$?
5. In the simulation part, whether the parameters selected have practical significance?
6. P6-P9. There are many writing mistakes in mathematical formulas, please check it.
7. In page 6, What are the parameters of levels of awareness σ and transmission β max based on ?
8. What are the advantages and disadvantages of the two models (ODE model and Network model) used in this paper?
9. The simulation results are not clearly expressed.

Review form: Reviewer 2 (Ana I. Bento)

Is the manuscript scientifically sound in its present form?

Yes

Are the interpretations and conclusions justified by the results?

Yes

Is the language acceptable?

Yes

Do you have any ethical concerns with this paper?

No

Have you any concerns about statistical analyses in this paper?

No

Recommendation?

Major revision is needed (please make suggestions in comments)

Comments to the Author(s)

Royal society open science review Mauricio SV paper

This is a timely study that addresses a well known limitation in dynamical transmission models. Incorporating dynamics behavior is therefore important especially in the context of an ongoing epidemic

There are things that I think could potentially improve the study

Discerning between ind that don't cooperate because their behavior is not elastic is very important so I would advise creating a scenario where this happens

Also having scenarios where information is lagged is important, as that will likely distort individuals' behavior and thus affect FOI. I recognize the authors have the parameter "awareness" but I do think it would be insightful to play with lags/ frictions in both awareness and compliance.

In infectious diseases models, "free riding" is difficult to extrapolate, precisely because of the moving target that is herd immunity. So I do think that perhaps the pay off should at some point be decomposed (not saying necessarily for this ms).

This is particularly important, with the need to discern between ind that don't cooperate because their behavior is not elastic vs those who have a choice. This is very important so I would advise creating a scenario where this happens, or at least devote some space to this in the discussion.

Explore different shapes in the relationship between $\beta(c)$ and $f(c)$ is also important (although I recognize not trivial). But as a theoretical exercise could give insights into the effects of payoffs in the context of the network topologies. These will no doubt affect the steady state as will lags and friction mentioned above.

Also while I understand that this will be beyond the current scope, I do think this problem should at some point be explored in a hierarchical network to explore the global vs local ideas already mentioned in the ms, to explore spreading across scales. As the authors mention not surprisingly steady state is affected by network structure!

I agree that there is no need for a BD process in this design.

Lastly, for future work I would advise "playing" with different games and experiment with hybrid strategies.

Decision letter (RSOS-210803.R0)

Dear Mr Cascante Vega,

The Editors assigned to your paper RSOS-210803 "How disease risk awareness modulates transmission: coupling infectious disease" have now received comments from reviewers and would like you to revise the paper in accordance with the reviewer comments and any comments from the Editors. Please note this decision does not guarantee eventual acceptance.

Please submit your revised manuscript and required files (see below) no later than 21 days from today's (ie 16-Aug-2021) date. Note: the ScholarOne system will 'lock' if submission of the revision is attempted 21 or more days after the deadline. If you do not think you will be able to meet this deadline please contact the editorial office immediately.

on behalf of Dr Oliver Schülke (Associate Editor) and Pete Smith (Subject Editor)
openscience@royalsociety.org

Associate Editor Comments to Author (Dr Oliver Schülke):

Dear Dr. Cascante Vega,

After having a VERY hard time finding reviewers, I now received comments of two experts. They both agree that your study is highly welcome and well conceived. Both make suggestions for improvements of the manuscript and for additional parameters to model to improve the reach of the study. I will be happy to receive a revised version of your manuscript at your earliest convenience.

With kind regards,
Oliver Schülke

Reviewer comments to Author:

Reviewer: 1

Comments to the Author(s)

In this paper, a coupling transmission dynamics with behavioral dynamics due to infection awareness is constructed and the dynamics of social behavior is modeled by game theory. The results show that as individuals increase their awareness the steady-state value of the final fraction of infected individuals in SIS model decreases. They extend the results to a spatial framework, incorporating a spatially-defined social network and show that even when individuals increase their awareness of the disease, the spatial structure itself defines the steady-state. The idea of this article is novel. The problems of this paper are as follows:

1. In the introduction of P 2, the ninth line of the first paragraph and the eighth line of the second paragraph, there are some writing mistakes, please check it.
2. In the Introduction part, the authors mentioned some work on mathematical modelling on both epidemiological and behavior change. However, some related work needs to be included: Linking the disease transmission to information dissemination dynamics: An insight from a multi-scale model study, *Journal of Theoretical Biology*, 2021; Markov-based solution for information diffusion on adaptive social networks, *Applied Mathematics and Computation*, 2020; Coupling dynamics of epidemic spreading and information diffusion on complex networks, *Applied Mathematics and Computation*, 2018; Evaluating risks using simulated annealing and Building Information Modeling, *Applied Mathematical Modelling*, 2015.
3. Can the derivation of the equation describing the behavior change be more detailed in the Eq.2.4 of page 4?
4. In the Network Model of P5, the line 7 of the first paragraph, $Pr = 1/\gamma$, should it be $Pr = \gamma$?
5. In the simulation part, whether the parameters selected have practical significance?
6. P6-P9. There are many writing mistakes in mathematical formulas, please check it.
7. In page 6, What are the parameters of levels of awareness σ and transmission β max based on?
8. What are the advantages and disadvantages of the two models (ODE model and Network model) used in this paper?
9. The simulation results are not clearly expressed.

Reviewer: 2

Comments to the Author(s)

Royal society open science review Mauricio SV paper

This is a timely study that addresses a well known limitation in dynamical transmission models. Incorporating dynamics behavior is therefore important especially in the context of an ongoing epidemic

There are things that I think could potentially improve the study

Discerning between ind that don't cooperate because their behavior is not elastic is very important so I would advise creating a scenario where this happens

Also having scenarios where information is lagged is important, as that will likely distort individuals' behavior and thus affect FOI. I recognize the authors have the parameter "awareness" but I do think it would be insightful to play with lags/ frictions in both awareness and compliance.

In infectious diseases models, “free riding” is difficult to extrapolate, precisely because of the moving target that is herd immunity. So I do think that perhaps the pay off should at some point be decomposed (not saying necessarily for this ms).

This is particularly important, with the need to discern between ind that don't cooperate because their behavior is not elastic vs those who have a choice. This is very important so I would advise creating a scenario where this happens, or at least devote some space to this in the discussion.

Explore different shapes in the relationship between $\beta(c)$ and $f(c)$ is also important (although I recognize not trivial). But as a theoretical exercise could give insights into the effects of payoffs in the context of the network topologies. These will no doubt affect the steady state as will lags and friction mentioned above.

Also while I understand that this will be beyond the current scope, I do think this problem should at some point be explored in a hierarchical network to explore the global vs local ideas already mentioned in the ms, to explore spreading across scales. As the authors mention not surprisingly steady state is affected by network structure!

I agree that there is no need for a BD process in this design.

Lastly, for future work I would advise “playing” with different games and experiment with hybrid strategies.

===PREPARING YOUR MANUSCRIPT===

Your revised paper should include the changes requested by the referees and Editors of your manuscript. You should provide two versions of this manuscript and both versions must be provided in an editable format:
 one version identifying all the changes that have been made (for instance, in coloured highlight, in bold text, or tracked changes);
 a 'clean' version of the new manuscript that incorporates the changes made, but does not highlight them. This version will be used for typesetting if your manuscript is accepted.

===PREPARING YOUR REVISION IN SCHOLARONE===

Author's Response to Decision Letter for (RSOS-210803.R0)

See Appendix A.

RSOS-210803.R1 (Revision)

Review form: Reviewer 1

Is the manuscript scientifically sound in its present form?

Yes

Are the interpretations and conclusions justified by the results?

Yes

Is the language acceptable?

Yes

Do you have any ethical concerns with this paper?

No

Have you any concerns about statistical analyses in this paper?

No

Recommendation?

Accept as is

Comments to the Author(s)

The work is well written in the present form.

Decision letter (RSOS-210803.R1)

Dear Mr Cascante Vega,

It is a pleasure to accept your manuscript entitled "How disease risk awareness modulates transmission: coupling infectious disease" in its current form for publication in Royal Society

Open Science. The comments of the reviewer(s) who reviewed your manuscript are included at the foot of this letter.

on behalf of Dr Oliver Schülke (Associate Editor) and Pete Smith (Subject Editor)
openscience@royalsociety.org

Reviewer comments to Author:
Reviewer: 1
Comments to the Author(s)
The work is well written in the present form.

Appendix A

Royal Society Open Science - Decision on Manuscript ID
RSOS-210803

Deadline resubmission: Please submit your revised manuscript and required files (see below) no later than 21 days from today's (ie 16-Aug-2021) date. Note: the ScholarOne system will 'lock' if submission of the revision is attempted 21 or more days after the deadline. If you do not think you will be able to meet this deadline please contact the editorial office immediately.

Editor Response:

Dear Dr. Cascante Vega,

After having a VERY hard time finding reviewers, I now received comments of two experts. They both agree that your study is highly welcome and well conceived. Both make suggestions for improvements of the manuscript and for additional parameters to model to improve the reach of the study. I will be happy to receive a revised version of your manuscript at your earliest convenience.

With kind regards,
Oliver Schülke

Reviews.

Reviewer 1 (Comments to the Author(s)):

In this paper, a coupling transmission dynamics with behavioral dynamics due to infection awareness is constructed and the dynamics of social behavior is modeled by game theory. The results show that as individuals increase their awareness the steady-state value of the final fraction of infected individuals in SIS model decreases. They extend the results to a spatial framework, incorporating a spatially-defined social network and show that even when individuals increase their awareness of the disease, the spatial structure itself defines the steady-state. The idea of this article is novel. The problems of this paper are as follows:

1. In the introduction of P 2, the ninth line of the first paragraph and the eighth line of the second paragraph, there are some writing mistakes, please check it.

Thanks for pointing out these writing issues. We have corrected them in the new version of the manuscript.

2. In the Introduction part, the authors mentioned some work on mathematical modelling on both epidemiological and behavior change. However, some related work needs to be included: Linking the disease transmission to information dissemination dynamics: An insight from a multi-scale model study, *Journal of Theoretical Biology*, 2021; Markov-based solution for information diffusion on adaptive social networks, *Applied Mathematics and Computation*, 2020; Coupling dynamics of epidemic spreading and information diffusion on complex networks, *Applied Mathematics and Computation*, 2018; Evaluating risks using simulated annealing and Building Information Modeling, *Applied Mathematical Modelling*, 2015.

Thanks for suggesting this interesting research papers, now we have revised them and included them in the introduction of the manuscript. These were included at the end of paragraph 2.

Epidemiological models have also been proposed for studying information diffusion on adaptive social networks [17,18] and have also been coupled with information dissemination dynamics to link disease transmission [19,20].

Citations to the commented articles correspond to 17-20

3. Can the derivation of the equation describing the behavior change be more detailed in the Eq.2.4 of page 4?

Thanks for the comments we agree that our derivation lacks some detail. We have included in the first section of the Supplementary Information the reduction of the 2 dimensional replicator dynamics system to the 1D equation presented in Eq. 2.4.

4. In the Network Model of P5, the line 7 of the first paragraph, $P_r = 1/\gamma$, should it be $P_r = \gamma$?

Thank you for noting this typo. Indeed, γ makes reference to the recovery rate of an individual, therefore $1/\gamma$ is defined as the recovery period (in days). The model is integrated using a daily time-step $\Delta t=1$. Hence, the probability of recovery P_r is given by $\Delta t \cdot 1/\gamma$. We corrected this in the Network model section in Materials and Methods.

We implemented a SIS model on a network assuming now susceptible individuals get infected by their infected neighbors (individuals with common edges) by a infection probability β_i and recover from the disease with the rate γ , same as used in the ODEs model Eq. 2.4. We integrate the models using a daily time-step of $\Delta t = 1$. Then each individual i has a state in the disease named Susceptible S or Infected I and the transition between those states is as described by Eq. 2.7.

5. In the simulation part, whether the parameters selected have practical significance?

Thank you for this thoughtful comment, our model considers a game-theory framework based where individuals change their behavior in pair-wise encounters. Parametrizing these models might be a challenge in the ongoing epidemic due to the huge uncertainty in modeling real-time people behavior. Other methods should be explored in order to account for these dynamical networks. This was included in the discussion at the end of paragraph 5.

Although, We choose the infectious period of the host as 7 days - 1 week, which follows the reported recovery rate range for different diseases for example as SARS [cite], COVID19 [cite]. Our framework its meant to be flexible and represent We then choose the nominal transmission probability values β_{max} to result in a basic reproductive number R_0 in three transmission regimes $R_0 \in \{2.2, 4.2, 6.3\}$ that represent normal, high and super high transmission settings. Different pathogens therefore are mapped in these levels of transmission. For example SARS R_0 was estimated to be in the range 2.2 to 3.6 (normal transmission)¹, SARS-CoV2 R_0 estimates presented early in the COVID19 pandemic present mean values from 2.5 - 6.1, but reconciled estimates accounting for differences in growth rates and generation intervals present values from 2.3 - 4.8 (high transmission)². And finally for Measles R_0 is generally cited to be in the range 12-18, however different estimates considering prior population immunity by vaccination estimate it in the range 6 - 9.5 (High transmission)³⁻⁵.

Thank you so much for the suggestion, we include this in the results and discussion section.

6. P6-P9. There are many writing mistakes in mathematical formulas, please check it.

Thanks for making us notice these mathematical typos, now we have corrected them and changed some of the notation to make everything easier to follow.

7. In page 6, What are the parameters of levels of awareness σ and transmission β_{max} based on.

The parameter β_{max} reflects the contact rate in the ODEs model and the infection probability given a contact with an infected individual. Therefore, it reflects the

transmissibility of the pathogen. The parameter awareness σ is however not linked to a specific measured quantity. We also think a future improvement in the work would be to include real world measures of this parameter. We have a couple of sentences mentioning these limitations. Importantly, our approach assume that this parameter somehow is capturing two features about the epidemic:

- 1. How the information about an outbreak and about the pathogen transmission is acquired by individuals in the populations.**
- 2. How individuals “care” or are aware about the epidemic and therefore how their perceived payoff changes.**

8. What are the advantages and disadvantages of the two models (ODE model and Network model) used in this paper?

Thanks for making us notice we did not discuss this. Now we have included it in the discussion section at paragraph 5.

For the ODEs equation model, as is broadly discussed in the infectious diseases modeling, the basic assumption that the FOI is proportional to the contact rate β_{max} the Susceptible density of the population S/N and the number of infected Individuals I is tied to an homogeneous mixing population (i.e. random encounters between any pairs of individuals). However, as is discussed in the manuscript, heterogeneities caused by both mobility of individuals across communities, spatial segregation principally by socio-economic factors. Therefore, these models often are discussed to reflect poorly on these heterogeneities, however they are often preferred also for capturing in few parameters and dynamics the complexities of infectious diseases dynamics. Even when the ODEs model relies on a huge assumption around the contact homogeneity in the population it can be used to model the real world population (millions of individuals). However the network models, as is an agent based model (ABMs) with a specific contact network, needs to update each agent state in each time step. In consequence, it's running time performance is poor for large populations. In this work we only use a toy network with 1000 individuals. Moreover, as epidemics unfold across populations public health non-pharmaceutical interventions (NPIs) are imposed, leading to a huge heterogeneity around interactions of people inside and outside their households and therefore varying the contact structure (social network) in time. In consequence, assuming a static contact network might reflect poorly real world settings. This however can be further improved as an understanding of local social networks and/or data collection about individuals movement is improved.

9. The simulation results are not clearly expressed.

Thanks for pointing this out. We have included a broad discussion of the simulation results.

Reviewer 2 (Comments to the Author(s)):

This is a timely study that addresses a well known limitation in dynamical transmission models. Incorporating dynamic behavior is therefore important especially in the context of an ongoing epidemic.

There are things that I think could potentially improve the study

1. Discerning between ind that don't cooperate because their behavior is not elastic is very important so I would advise creating a scenario where this happens.

Thanks for pointing this out. We only use the Scale-Free network model to include this non-elastic behavior of individuals. We updated some simulations as in Figure 3 of the main text where we assume:

1. A random 10% of individuals do not cooperate.
2. A random 25% of individuals do not cooperate.
3. A random 50% of individuals do not cooperate.
4. A random 10% of individuals always cooperate.
5. A random 25% of individuals always cooperate.
6. A random 50% of individuals always cooperate.

Now we have added the simulations to the supplementary information and updated the discussion correspondingly

Figure Effect of non-elastic behavior over disease dynamics for the global information scheme. Infected fraction time dynamics for different cooperative scenarios. All simulations begin by randomly initializing half of the population as cooperators (and defectors). From these initial behavior states, we then fixed the 10%, 25%, and 50% of the initialized individuals to follow a fixed strategy (cooperate or defect always). We did this for different combinations of R_0 and awareness σ . From top row to bottom row awareness σ increases from low, medium, and high awareness as indicated in the methods section. From left to right transmission intensity increases from the normal, high, and high transmission as indicated in the methods section.

Figure Effect of non-elastic behavior over social dynamics for the global information scheme. Cooperation fraction time dynamics for different cooperative scenarios. All simulations begin by randomly initializing half of the population as cooperators (and defectors). From this initial behavior states, we then fixed the 10%, 25%, and 50% of the initialized individuals to follow a fixed strategy (cooperate or defect always). We did this for different combinations of R_0 and awareness σ . From top row to bottom row awareness σ increases from low, medium and high awareness as indicated in the methods section. From left to right transmission intensity increases from normal, high and super high transmission as indicated in the methods section.

Figure Effect of non-elastic behavior over disease dynamics for the local information scheme. Infected fraction time dynamics for different cooperative scenarios. All simulations begin by randomly initializing half of the population as cooperators (and defectors). From this initial behavior states, we then fixed the 10%, 25%, and 50% of the initialized individuals to follow a fixed strategy (cooperate or defect always). We did this for different combinations of R_0 and awareness σ . From top row to bottom row awareness σ increases from low, medium and high awareness as indicated in the methods section. From left to right transmission intensity increases from normal, high and super high transmission as indicated in the methods section.

Figure Effect of non-elastic behavior over social dynamics for the local information scheme. Cooperation fraction time dynamics for different cooperative scenarios. All simulations begin by randomly initializing half of the population as cooperators (and defectors). From these initial behavior states, we then fixed the 10%, 25%, and 50% of the initialized individuals to follow a fixed strategy (cooperate or defect always). We did this for different combinations of R_0 and awareness σ . From top row to bottom row awareness σ increases from low, medium and high awareness as indicated in the methods section. From left to right transmission intensity increases from normal, high and super high transmission as indicated in the methods section.

2. Also having scenarios where information is lagged is important, as that will likely distort individuals' behavior and thus affect FOI. I recognize the authors have the parameter "awareness" but I do think it would be insightful to play with lags/ frictions in both awareness and compliance.

Thanks for this comment. We also believe that incorporating a mechanism in the model behind potential lags in how individuals access information about the disease risk awareness as stated in the Methods section might result in decoupling in the game and epidemiological dynamics. This will also reflect the reporting process of infectious disease data, usually incident cases, more realistic. However, we would like to keep this for future work where more realistic models of both disease and social dynamics are included as we discuss in the following comments.

3. In infectious diseases models, "free riding" is difficult to extrapolate, precisely because of the moving target that is herd immunity. So I do think

that perhaps the pay off should at some point be decomposed (not saying necessarily for this ms).

Thanks for this comment. While free-riding might affect the transient dynamics in susceptible - infected - susceptible epidemiological (SIS) models, as we are characterizing the system with the steady-state we argue this effect is neglectable. Moreover, herd immunity is never achieved in an SIS model as the hosts do not develop any immunity against the pathogen. However, we do believe extending this framework to different and more complex epidemiological dynamics should be done and characterized carefully in future works considering the effect of herd immunity in the population and individual-level behavior. As you also advise later, increasing the complexity in the epidemiological model should be carefully coupled with the implementation of different games in the social dynamics.

4. This is particularly important, with the need to discern between ind that don't cooperate because their behavior is not elastic vs those who have a choice. This is very important so I would advise creating a scenario where this happens, or at least devote some space to this in the discussion.

Thanks for these suggestions. Now following the reviewer suggestion in the first comment we have run different simulations assuming a random set of individuals with not elastic behavior during all the simulation indicated and update the discussion.

5. Explore different shapes in the relationship between $\beta(c)$ and $f(c)$ is also important (although I recognize not trivial). But as a theoretical exercise could give insights into the effects of payoffs in the context of the network topologies. These will no doubt affect the steady state as will lags and friction mentioned above.

Thanks for pointing this out. We also thought about this, and therefore it is interesting that you have it in mind too. We have now included other functional responses and added them to the manuscript. However, as we already know that the dynamics will only be affected in the ODEs model and the scale-free Network model (as discussed in the manuscript) we only study this with different functional responses between $\beta(c)$ and $f(c)$

For studying the relation of different shapes between of $\beta(c)$ and c (the fraction of cooperators) is the independent variable $\beta(c) = f(c)$, we parametrize the function with two values: β_{max} and β_{min} , the maximum and minimum contact rate or probability of transmission (ODE and network models respectively). As our first rationale was choosing $\beta(c) = \beta_{max} \exp(-c)$, for having the same minimum and maximum $\beta(c)$ we choose $\beta_{min} = \beta_{max} \exp(-1)$ regardless of the chosen functional

response. We then vary β_{max} , as explained in the Methods Section. We then consider five (5) types of functional responses: Convex, Exponential (original), Linear, Concave 1 and Concave 2 as shown in the figure below.

Figure x Functional responses of transmission rate beta and cooperator fraction (c). Each line represents a different functional response of the transmission rate or transmission probability as a function of the fraction of cooperators in the population. We plot the functional responses choosing $\beta_{max} = 0.3$. We fixed the minimum value as $\beta_{min} = \beta_{max} \exp(-1)$. This choice of β_{min} follows the rationale that assuming all individuals cooperate $c = 1$ the exponential functional response $\beta(c) = \beta_{max} \exp(-c)$ results in the minimum value of the transmission probability we consider β_{min} .

We then study the steady state dynamics of the system as shown in Figure 2 of the main manuscript. We study the steady state dynamics in both ODEs and Network models as shown in Figures below.

Figure x. Steady state solution of population infected fraction (top), and cooperators fraction (bottom) in global information scheme. Top row: Infected fraction at steady state. Bottom row: Cooperator fraction at steady state. From left to right heat maps represent the steady states changing the functional response of the transmission rate: Convex, exponential, linear, and both concave functions. Color codes are described in the colorbar

Figure. Steady state solution of population infected fraction (top), and cooperators fraction (bottom) in global information scheme. Top row: Infected fraction at steady state. Bottom row: Cooperator fraction at steady state. From left to right heat maps represent the steady states changing the functional response of the transmission rate: Convex, exponential, linear, and both concave functions. Color codes are described in the colorbar. simulations were conducted over a 1000 node scale-free network.

Figure. Steady state solution of population infected fraction (top), and cooperators fraction (bottom) in local information scheme. Top row: Infected fraction at steady state. Bottom row: Cooperator fraction at steady state. From left to right heat maps represent the steady states changing the functional response of the transmission rate: Convex, exponential, linear, and both concave functions. Color codes are described in the colorbar. Simulations were conducted over a 1000 node scale-free network.

We then characterize the time dynamics by considering three (3) levels of transmission $R_0 \in \{2.1, 4.2, 6.3\}$ named normal, high and super high transmission to represent different pathogens regimes. Furthermore by considering three levels of population-individuals awareness $\sigma \in \{0.5, 0.7, 1\}$ low, medium and full as described in Methods section we study the time dynamics as shown in Figures below for the infected and cooperators fraction in the population. Results are shown for both the ODEs and Network models.

Figure. Infected fraction dynamics changing functional responses in the ODEs model. Infected fraction time dynamics for different functional responses for $\beta(c)$. The solid lines represent the mean value of 100 simulations and colors are indicated in the legend. From top row to bottom row awareness σ increases from low, medium and high awareness as indicated in the methods section. From left to right transmission intensity increases from normal, high and super high transmission as indicated in the methods section.

Figure. Infected fraction time dynamics for different functional responses for $\beta(c)$. The solid lines represent the mean value of 100 simulations and colors are indicated in the legend. From top row to bottom row awareness σ increases from low, medium and high awareness as indicated in the methods section. From left to right transmission intensity increases from normal, high and super high transmission as indicated in the methods section.

Figure. Infected fraction dynamics changing functional responses in the Network model. Infected fraction time dynamics for different functional responses for $\beta(c)$. The solid lines represent the mean value of 100 simulations and colors are indicated in the legend. From top row to bottom row awareness sigma increases from low, medium and high awareness as indicated in the methods section. From left to right transmission intensity increases from low, medium and high transmission as indicated in the methods section.

Figure. Cooperation fraction dynamics changing functional responses in the Network model. Cooperators fraction time dynamics for different functional responses for $\beta(c)$. The solid lines represent the mean value of 100 simulations and colors are indicated in the legend. From top to bottom awareness σ increases from low, medium and high awareness as indicated in the methods section. From left to right transmission intensity increases from low, medium and high transmission as indicated in the methods section.

We found that the steady state dynamics of both the epidemic and social dynamics are affected by the type of functional response $\beta(c)$ chosen.

6. Also while I understand that this will be beyond the current scope, I do think this problem should at some point be explored in a hierarchical network to explore the global vs local ideas already mentioned in the ms, to explore spreading across scales. As the authors mention, not surprisingly steady state is affected by network structure!

Thanks again for this comment. We also believe that the network structure defines the steady state dynamics of the system. Now we have aggregated this in the discussion and we have mentioned that for future works we necessarily should include simulations in hierarchical networks.

7. I agree that there is no need for a BD process in this design.

Thanks for the comment

8. Lastly, for future work I would advise “playing” with different games and experiment with hybrid strategies.

Thanks for pointing this out, we are currently working on retention of this framework, where we are increasing the complexity in both epidemiological models (including other stages of the diseases) and changing the game dynamics using other archetypical games as the cooperation game.

References

1. Lipsitch, M. *et al.* Transmission Dynamics and Control of Severe Acute Respiratory Syndrome. *Science* **300**, 1966–1970 (2003).
2. Park, S. W. *et al.* Reconciling early-outbreak estimates of the basic reproductive number and its uncertainty: framework and applications to the novel coronavirus (SARS-CoV-2) outbreak. *12*.
3. Guerra, F. M. *et al.* The basic reproduction number (R_0) of measles: a systematic review. *Lancet Infect. Dis.* **17**, e420–e428 (2017).
4. Rubió, P. P. Is the basic reproductive number (R_0) for measles viruses observed in recent outbreaks lower than in the pre-vaccination era? *Eurosurveillance* **17**, 20233 (2012).
5. Vivancos, R. *et al.* An ongoing large outbreak of measles in Merseyside, England, January to June 2012. *Eurosurveillance* **17**, 20226 (2012).